# Mycobiota and Mycotoxin Contamination of Traditional and Industrial Dry-Fermented Sausage *Kulen*

**DOI:** 10.3390/toxins13110798

**Published:** 2021-11-12

**Authors:** Tina Lešić, Manuela Zadravec, Nevijo Zdolec, Ana Vulić, Irena Perković, Mario Škrivanko, Nina Kudumija, Željko Jakopović, Jelka Pleadin

**Affiliations:** 1Laboratory for Analytical Chemistry, Croatian Veterinary Institute, Savska Cesta 143, 10000 Zagreb, Croatia; lesic@veinst.hr (T.L.); vulic@veinst.hr (A.V.); kudumija@veinst.hr (N.K.); 2Laboratory for Feed Microbiology, Croatian Veterinary Institute, Savska Cesta 143, 10000 Zagreb, Croatia; zadravec@veinst.hr; 3Department of Hygiene, Technology and Food Safety, Faculty of Veterinary Medicine, University of Zagreb, Heinzelova 55, 10000 Zagreb, Croatia; nzdolec@vef.unizg.hr; 4Croatian Veterinary Institute, Regional Veterinary Institute Vinkovci, Ul. Josipa Kozarca 24, 32100 Vinkovci, Croatia; perkovicirena1512@gmail.com (I.P.); skrivanko@veinst.hr (M.Š.); 5Faculty of Food Technology and Biotechnology, University of Zagreb, Pierottijeva 6, 10000 Zagreb, Croatia; zjakopovic@pbf.hr

**Keywords:** meat products, cyclopiazonic acid, aflatoxin B_1_, ochratoxin A, LC-MS/MS, moulds, *Penicillium*, *Aspergillus*, molecular identification

## Abstract

The aim of this study was to identify and compare surface mycobiota of traditional and industrial Croatian dry-fermented sausage *Kulen*, especially toxicogenic species, and to detect contamination with mycotoxins recognized as the most important for meat products. Identification of mould species was performed by sequence analysis of beta- tubulin and calmodulin gene, while the determination of mycotoxins aflatoxin B_1_ (AFB_1_), ochratoxin A (OTA), and cyclopiazonic acid (CPA) was carried out using the LC-MS/MS (liquid chromatography-tandem mass spectrometry) method. The results showed a significantly higher number of mould isolates and greater species (including of those mycotoxigenic) diversity in traditional *Kulen* samples in comparison with the industrial ones. *P. commune*, as a potential CPA-producer, was the most represented in traditional *Kulen* (19.0%), followed by *P. solitum* (16.6%), which was the most represented in industrial *Kulen* samples (23.8%). The results also showed that 69% of the traditional sausage samples were contaminated with either CPA or OTA in concentrations of up to 13.35 µg/kg and 6.95 µg/kg, respectively, while in the industrial samples only OTA was detected (in a single sample in the concentration of 0.42 µg/kg). Mycotoxin AFB_1_ and its producers were not detected in any of the analysed samples (<LOD).

## 1. Introduction

Fermented meat products are the most popular group of meat products produced using both traditional and industrial technologies. The production of traditional and industrial Croatian dry-fermented sausage *Kulen* differs in recipe, technology, and environmental conditions, the industrial production, thereby using starter cultures and additives in controlled and standardised settings [1]. Production environments affect the specificity and the density of mycobiota that overgrow the surface of dry-fermented meat products during ripening, and consequently also the quality and safety of these products [2,3]. Although the presence of surface moulds is generally appreciated because of their contribution to the development of characteristic dry-fermented product flavour and quality, in cases of uncontrolled mould growth unwanted species develop, causing unpleasant sensory characteristics or adverse consumer health effects. Some species of the *Penicillum* and the *Aspergillus* genus that usually overgrow dry-fermented meat products are capable of producing mycotoxins [4,5,6,7,8]. Surface inoculation with selected fungal starter cultures helps minimising the competition with toxigenic moulds and enhances ripening, hence cutting the chance of potential mycotoxin contamination to the minimum [9].

Mycotoxins are toxic secondary mould metabolites, globally recognised as food safety hazards responsible for acute and chronic toxicity [10]. Due to their common appearance in dry-fermented meat products, as well as their toxicity, mycotoxins of the highest interest are ochratoxin A (OTA), aflatoxin B_1_ (AFB_1_), and cyclopiazonic acid (CPA) [5,11,12]. OTA is classified by the International Agency for Research on Cancer (IARC) as a Group 2B possible human carcinogen and has been found to be the dominating meat products’ contaminant [6,13,14,15], while AFB_1_ belongs to the IARC Group 1 of proven carcinogens and occurs less frequently in meat products and in lower concentrations [7,16,17]. CPA is considered to be a potentially dangerous mycotoxin that can damage the digestive organs, the myocardium, and the skeletal muscles, but due to the insufficiently relevant carcinogenicity studies, the IARC did not classify it yet [18]. Data on the occurrence of CPA in different meat products are very limited, although some authors report this occurrence to be substantial [12,19]. To the best of our knowledge, as of now, only one study investigated the CPA contamination of traditional Croatian dry-fermented meat products [20], while the investigation into the industrial products hasn’t been done yet, while the studies carried out in other European countries are very few in number [21,22,23].

Although dry-fermented sausages’ manufacturing includes washing, brushing, and packaging that enables the removal of moulds, studies showed that the toxigenic mould population that grows on the surface of dry-fermented meat products makes the largest contribution to the overall mycotoxin contamination of the product. As for CPA, *P. commune* has been described as the main source of contamination of meat products. Species responsible for OTA contamination of meat products are mostly *P. verrucosum* and *P. nordicum*, while the presence of AFB_1_ in dry-fermented meat products is the result of *A. flavus* and *A. parasiticus* activity [2,5,19]. Some *A. flavus* strains can produce not only aflatoxins, but CPA as well, rendering the possibility of their co-occurrence. Therefore, the 2006 Panel on Contaminants in the Food Chain (CONTAM) stressed the importance of the assessment of simultaneous exposure to both mycotoxins [18,24]. Besides direct mould contamination, mycotoxins can be present in meat products due to the naturally contaminated spice mixtures or the indirect transfer from farm animals exposed to naturally contaminated feed (carry-over effect) [25,26,27].

The aim of this study was to investigate and compare the surface mycobiota and AFB_1_, OTA, and CPA contamination of traditional and industrial *Kulen*, one of the protected autochthonous Croatian dry-fermented sausages. Particular attention was paid to toxicogenic species, which were put in relation to the determined mycotoxin levels.

## 2. Results and Discussion

### 2.1. LC-MS/MS Validation

Validation results, together with the LOD and the LOQ, are shown in Table 1. Calibration curves pertinent to each of the mycotoxins showed good linearity, with the regression coefficient of R2 = 0.998 for AFB_1_, 0.999 for OTA, and 0.999 for CPA in the calibration range of 0.05 to 1 ng/mL for AFB_1_, 0.2 to 5 ng/mL for OTA, and 0.5 to 25 ng/mL for CPA. The average recoveries ranged from 91.4% to 119.4%, which is consistent with the requirement for the method trueness at ≤1 μg/kg level (−50% to +20%), expressed as the recovery [28]. Evaluation of the matrix effect showed a slight ion enhancement effect in the 1.8–6.4% range. Validation results demonstrate the suitability of the method used for the investigation into the contamination of dry-fermented sausages (such as *Kulen*) with OTA, AFB_1_, and CPA.

### 2.2. Mycobiota in Traditional and Industrial Kulen

Mould species identified on the surface of traditional and industrial *Kulen* in this study, are shown in Table 2. The prevalence of the *Penicillium* in comparison with the *Aspergillus* genus was higher in both *Kulen* types, 79% of the identified moulds thereby being of the *Penicillium*, and 21% of the *Aspergillus* genus. This agrees with the previous research on dry-fermented meat products, especially dry-fermented sausages, and with the facts that most of the *Penicillium* species are psychrotolerant and psychrophilic and that the ripening of *Kulen* mostly takes place during winter [6,29]. Alapont et al. [5] pointed out that the *Penicillium* genus predominated in samples taken in ripening chambers, indicating that the air inside them may be an important source of contamination. On the surfaces of both *Kulen* types, a total of 4 *Aspergillus* and 10 *Penicillium* species was identified.

In comparison with the industrial *Kulen* samples, the traditional *Kulen* samples harboured a significantly higher number of more diversified mould isolates (*p* = 0.022), which can be attributed to uncontrolled production and seasonal variation seen in a traditional line of work as compared to industrial production. In addition to ripening conditions, the diversity of species colonising traditional and industrial *Kulen* can also be attributed to different ripening chambers’ surroundings from which mould spores mostly penetrate into the chamber [2]. Perrone et al. [9] pointed out that bio-competition in terms of application of desirable protective cultures on the surface of dry-fermented meat products is widely investigated and attempted to be implemented in industrial settings.

From traditionally produced *Kulen*, a total of 42 mould isolates, out of which 33 of the *Penicillium,* and nine of the *Aspergillus* genus, belonging to 13 mould species, were retrieved, while from industrial *Kulen* 21 mould isolates, out of which 17 of the *Penicillium* and four of the *Aspergillus* genus, belonging to only eight mould species, were recovered. *P. atramentosum* was found only on the surfaces of industrial *Kulen* samples, while six species (*P. polonicum* and *P. brevicompactum* in more than one isolate and *A. tubingensis, A. westerdijkiae, P. nordicum,* and *P. nalgiovense* in only one isolate) were found only on the surfaces of traditional *Kulen* samples.

The most represented species found in traditional *Kulen* samples (hereby reported together with the pertaining relative densities, Dr = the number of given species isolates/total number of isolates × 100, reported based on the *Kulen* type) were *P. commune* (Dr = 19.0%), *P. solitum* (Dr = 16.6%), and *A. pseudoglaucus* (Dr = 14.3%), while the species most commonly encountered in industrial *Kulen* samples were *P. solitum* (Dr = 23.8%), *P. corylophilum* (Dr = 19.1%) and *P. citrinum* (Dr = 14.3%). *P. commune* and *P. solitum* are close relatives often dominating in fermented sausages, *P. solitum* thereby not producing mycotoxins, and *P. commune* producing CPA [30]. Earlier data show that, in most cases, *P. commune* and *P. solitum* are vastly represented in the sausage surface mycoflora, right after *P. nalgiovense* and *P. chrysogenum* [6,29,31,32]. *P. nalgiovense* is usually selected as a commercial starter culture, and in this study, *P. nalgiovense* was detected in only one traditional *Kulen* sample, whereas in earlier studies involving Croatian Slavonian *Kulen* and domestic sausages it wasn’t detected at all [6,7]. *P. corylophilum*, found to be predominant in industrial *Kulen* samples under this study, was also identified in the previous study of traditional Slavonian *Kulen* [7]. Contrary to 17 mould species identified in traditional *Kulen* in this study, the above research managed to identify only seven mould species [7]. The prevalence of *A. pseudoglaucus* can be explained by the fact that the ascospores of *Aspergillus* teleomorphs can survive a wide range of temperatures (4–43 °C), and very low water activity (0.71) up to 120 days. *A. pseudoglaucus* and *A. proliferans* were also predominating in different types of Croatian dry-fermented sausages [6,29]. *P. citrinum* is one of the most common fungal species on Earth, its conidiospores being one of the most common spores present in the air. Its ability to grow at a_w_ lower than 0.80 and the temperature of 5 °C helps this species to secure a niche in a wide range of habitats [33].

**Table 2 toxins-13-00798-t002:** Mycobiota recovered from the surfaces of traditional and industrial *Kulen*.

Genus	Species	Traditional *Kulen* (*n* = 16)		Industrial *Kulen* (*n* = 10)	
		N	Dr (%)	N	Dr (%)
*Aspergillus*	*A. pseudoglaucus*	6	14.3	2	9.5
	*A. proliferans*	1	2.4	2	9.5
	*A. tubingensis*	1	2.4	-	-
	*A. westerdijkiae* *	1	2.4	-	-
*Penicillium*	*P. solitum*	7	16.6	5	23.8
	*P. commune* **	8	19.0	1	4.8
	*P. corylophilum*	4	9.5	4	19.1
	*P. citrinum*	4	9.5	3	14.3
	*P. polonicum* **	5	11.9	-	-
	*P. sumatraense*	1	2.4	2	9.5
	*P. atramentosum*	-	-	2	9.5
	*P. brevicompactum*	2	4.8	-	-
	*P. nordicum* *	1	2.4	-	-
	*P. nalgiovense*	1	2.4	-	-
Total N of isolates		42		21	

N = number of isolates; Dr (relative density) = number of isolates of a given species/total number of isolates × 100, displayed based on the *Kulen* type; * OTA producers; ** CPA producers [30,34].

### 2.3. Mycotoxins in Traditional and Industrial Kulen

This study investigated the contamination of *Kulen* with mycotoxins most important for meat products and includes the comparison between mycotoxin concentrations in traditional and industrial samples. Mycotoxin concentrations determined in *Kulen* samples under this study are shown in Table 3. The results showed that 69% of the traditional *Kulen* samples were contaminated with either CPA or OTA; as opposed to that, only one industrial *Kulen* sample was (OTA) contaminated. The co-occurrence of the two was not determined in any of the samples. Mycotoxin contamination is related to mould growth and conditioned by relative humidity, temperature, pH, and a_w_. For example, higher mycotoxin production is determined at aw 0.99 in comparison with that of 0.97 and 0.95, so that besides washing of product surface to prevent an extensive mould growth, optimal manufacturing conditions are key to mycotoxin contamination prevention [35].

In this study, mycotoxin AFB_1_ wasn’t detected in any of the analysed samples. This study failed to identify either the species known to produce AFB_1_ in dry-fermented meat products, such as *A. flavus* and *A. parasiticus*, or any other species possibly capable of such a production. As for other studies dealing with AFB_1_ meat products’ contamination, the recent research on different types of dry-fermented sausages by [15] also failed to find any aflatoxin AFB_1_, AFB_2_, AFG_1_, or AFG_2_ contamination, while some previous research on meat products reported AFB_1_ concentrations in the range of 0.92–3.0 µg/kg [6,17,36]. A higher average AFB_1_ concentration of 11.79 µg/kg was determined in Slavonian *Kulen* in cases when the product was neglected to be cleaned during production so that the surface moulds remained in place and grew unhindered [7]. Despite the absence of AFB_1_ contamination established by this study and its poor representation in meat products investigated by other studies, further investigation of this carcinogenic aflatoxin and regular controls using highly sensitive confirmatory methods should be performed, since mycotoxin contamination depends on various factors, some of which cannot be influenced, such as regional climatic conditions that change every year. For instance, a few years ago, due to the weather conditions, high AFB_1_ concentrations were determined in feed, not only in Croatia, but in other European countries, as well, suggesting the possibility of indirect contamination of foodstuffs of animal origin [37,38].

**Table 3 toxins-13-00798-t003:** Mycotoxin concentrations in traditional and industrial *Kulen*.

Mycotoxins		Traditional *Kulen* (*n* = 16)	Industrial *Kulen* (*n* = 10)
AFB_1_	Mean of positives * ± SD ((µg/kg)	<LOD	<LOD
	Min (µg/kg)	<LOD	<LOD
	Max (µg/kg)	<LOD	<LOD
OTA	Mean of positives * ± SD	2.16 ± 2.72	0.42 ± na
	*n*+/*n*+%	6/38	1/10
	Min (µg/kg)	<LOD	<LOD
	Max (µg/kg)	6.95	0.42
CPA	Mean of positives * ± SD (µg/kg)	7.39 ± 4.09	<LOD
	*n*+/*n*+%	5/31	0/0
	Min (µg/kg)	<LOD	<LOD
	Max (µg/kg)	13.35	<LOD

LOD—limit of detection, LOQ—limit of quantification; * mycotoxin concentration above the LOD; *n*+: number of positive samples; *n*+%: per- cent of positive samples; na—not applicable.

OTA was detected in 27% of the analysed *Kulen* samples, out of which 38% were traditional *Kulen* samples in which OTA concentration of up to 6.95 µg/kg was found. As opposed to that, only one industrial *Kulen* sample was OTA-contaminated, but in concentration below the LOQ. At the European level, no guidance value for OTA or other mycotoxins in meat and meat products hasn’t been established so far, while in Italy, the Ministry of Health set the guideline value of 1 µg/kg for pork meat and derived products [14,39]. In our research, OTA concentrations found in two traditional *Kulen* samples exceeded that limit (3.91 and 6.95 µg/kg, respectively).

*P. nordicum*, identified in this study in one sample in which OTA concentration of 0.32 µg/kg was established, is known to produce OTA and can grow on dry-fermented meat products with high protein and salt content at lower temperatures [6]. It has been shown that *P. nordicum* can produce OTA within a wide temperature range and under relatively stressing abiotic interacting conditions. OTA production at 0.85 a_w_ was reported, but the optimal conditions appear to be the temperature of 15 °C and the a_w_ of 0.90 [9]. Studies showed that *A. westerdijkiae* can also produce OTA, while the examined strains of *A. pallidofulvus*, which belongs to the same clade as *A. westerdijkiae* and *A. ochraceus*, were reported to be unable to do so [9,40]. In one sample under this study in which *A. westerdijkiae* was identified, OTA was not detected (Appendix A).

For *A. tubingensis* of the *A. Nigri* section, to which ochratoxigenic species *A. niger* and *A. carbonarius* belong, as well, conflicting results regarding its ability to produce OTA have been published during the recent years. In the research of Storari et al. [41], OTA was not detected in any of the *A. tubingensis* extracts but was easily detected in the ochratoxigenic species *A. niger*. Further investigation into the presence of not only ochratoxigenic strains and the presence/absence of OTA biosynthetic genes, but also of other toxigenic strains and mycotoxins, are required. Another limitation of this study is the number of analyzed samples, which is not large enough because of the small number of available Croatian *Kulen* producers, especially industrial ones. Mycotoxin synthesis occurs only under restrictive conditions so that the line of production doesn’t necessarily correlate with toxicogenic mould growth. Besides mould-mediated contamination, the established low-level OTA contamination could be consequential to the use of contaminated spices, such as red spicy peppers and garlic, or the use of contaminated raw meat [26,27,42]. Pleadin et al. [42] determined OTA concentrations in spices used in the production of Slavonian *Kulen* to rise up to 8.11 µg/kg in red paprika, while in raw meat OTA failed to be detected.

Low OTA levels found in this study can also be explained by the presence of casing, which, if intact, can act as a sausage protective system. The study by Rocanda et al. [14] showed that OTA was detected in 25% of the analysed casings in concentrations of up to 98.52 µg/kg, while the edible parts were contaminated only in three samples in levels above 1 µg/kg. In earlier studies of Croatian dry-fermented sausages, the determined OTA concentrations ranged from 0.14 µg/kg to 19.84 µg/kg in Slavonian *Kulen* whose surface was not brushed to remove overgrown moulds during ripening [6,7,15,17,36]. In dry-fermented sausages from other European countries, OTA concentrations were in the range of <LOD—18 µg/kg [14,43,44].

CPA was detected in 19% of the analysed *Kulen* samples, but only in traditional *Kulen* and in a concentration of up to 13.35 µg/kg. The occurrence of CPA in meat products is poorly investigated; one of the reasons behind that is its challenging analysis. Data published so far indicate a wide concentration range of CPA in dry-cured ham samples (36–540 µg/kg) [21]. One research on CPA occurrence in Croatian traditional dry-fermented sausages revealed its presence in 7 out of 47 analysed samples in the concentration of 2.55 µg/kg to 59.80 µg/kg [20]. As compared to the study by Vulić et al. [20], results from this study revealed similar CPA concentrations (2.50–13.35 µg/kg), except for one higher value of 59.80 µg/kg. CPA can accumulate in meat products during storage, as reported by [19] since it has been shown to be more stable than OTA and CIT.

CPA is consistently produced in food by *P. grieseofulvum, P. camemberti*, *P. commune*, *A. flavus*, *A. oryzae,* and *A. tamarii* [18]. Upon the inoculation of *P. griseofulvum* into dry-fermented sausages, a significant CPA production was witnessed at lower temperatures (i.e., 12 °C), increasing during production possibly due to the intense meat proteolysis and the subsequent release of free amino acids, among other tryptophan, a direct CPA precursor [9,22]. *P. griseofulvum* wasn’t detected in this study, but *P. commune* was the predominant species found in traditional *Kulen* samples. In the industrial *Kulen* samples, in which no CPA was detected, *P. commune* was identified in only one sample. The study of natural mycobiota of traditional Spanish dry-cured hams by [5] revealed that 33.7% of the isolated fungal strains produced CPA, out of which 66.6% were CPA-producing *P. commune* strains. The authors also reported that two out of five analysed strains of *P. polonicum*, shown to be highly toxigenic, were able to produce CPA. *P. atramentosum* and *P. solitum* isolated from dry-cured hams were also able to produce CPA, but in very low concentrations. In addition to that, these species were previously not reported as CPA-producing fungi, so that more evidence should be collected. In this study, moulds were identified using the ITS region, so that a misidentification in the database could occur. In five CPA-contaminated samples determined in this study, the identified species capable of CPA production were *P. commune* (in three samples) and *P. polonicum* (in the same number of samples).

It is worth mentioning that *P. citrinum*, a potential mycotoxigenic species that can produce citrinin (CIT), was found in 10% of samples under this study, both in traditional and industrial *Kulen*. The occurrence of citrinin was investigated in only one Croatian study of meat products using the ELISA method, which tagged it as an insignificant contaminant (present in 4.44% of samples) present in concentrations around the LOD of the analytical method (1.0–1.3 µg/kg) [36]. However, since *P. citrinum* is frequently isolated from dry-cured meat products and given that the production of CIT is also climatically conditioned, its occurrence certainly varies across the production years. Also, in view of the scarce evidence on CIT contamination and possible OTA and CIT co-occurrence described in literature, it would be of interest to further study CIT contamination of meat products using highly sensitive analytical methods [6,12,19,29]. Bailly et al. [19] reported relatively high amounts of citrinin produced in dry meat by toxigenic *P. citrinum* strain after 16-day incubation at 20 °C.

From the toxicity standpoint, investigations into mycotoxin occurrence are very important for the assessment of consumer exposure to these contaminants via meat products as one of the major constituents of the human diet. This goes especially for the investigations into CPA as a virtually unexplored mycotoxin. On top of that, research in this field is also of the essence for compiling legislation governing mycotoxin presence in meat and meat products, since no maximal or guidance values have insofar been established in European countries for this type of products.

## 3. Conclusions

The results revealed a significantly higher number of mould isolates and greater mould species (including mycotoxigenic species) diversity in traditional in comparison with industrial *Kulen* samples, which can be attributed to non-regulated and variable seasonal conditions under which traditional as opposed to industrial production takes place. A higher prevalence of the *Penicillium* genus in comparison with the *Aspergillus* genus can be attributed to the continental climate environment in which *Kulen* samples were produced, preferred by the *Penicillium* species. *P. commune*, a potential CPA producer, was the most represented species identified in traditional *Kulen*. The results showed that 69% of the traditional *Kulen* samples were contaminated with either CPA or OTA in higher concentrations, while in the industrial *Kulen* samples only OTA was detected in a single sample and lower concentration. AFB_1_ wasn’t detected in any of the analysed samples; the same goes for the moulds producing it. The study showed that traditional homemade *Kulen* can be overgrown by unwanted mould species during ripening and hence contaminated with mycotoxins. This highlights the importance of hygiene control during the production process. CPA contamination can be consequential to direct CPA production by isolated mould producers, such as *P. commune*. Since the potential OTA producers were found in only two samples, the detected fairly low OTA levels can probably be attributed to the use of contaminated spices or other raw materials. Further research should be focused on other types of traditional meat products, especially those longer ripen, as well as on other mycotoxins that can be found in dry-fermented meat products, such as citrinin.

## 4. Materials and Methods

### 4.1. Dry-Fermented Sausage Kulen Samples

Samples of *Kulen* were obtained during 2020 from 26 producers, 16 of them thereby using traditional and 10 using industrial production technologies. The sampling in the amount of 1.5–2.0 kg was carried out in full line with the Commission Regulation [45]. Traditional *Kulen* samples were produced in 2019/2020 by family farms seated in eastern Croatia (Slavonija and Baranja), where this sausage represents a traditional food brand, while industrial *Kulen* samples were produced in 2020 by different licensed industrial producers.

According to Regulation [46], *Kulen* belongs to the group of dry-fermented sausages enlisted within the category of non-thermally processed meat products. It must contain at least 22% of meat protein and not more than 40% of water. *Kulen* is produced from the first- or the second-category pork meat cut into small pieces and enriched with solid fat. The prepared mixture is chopped using an 8- to 10- mm hole meat grinder. After the addition of salt and spices (sweet and hot red pepper, garlic, black pepper), the minced meat is stuffed into the pig appendix (*intestinum caecum*). After that, sausages are dried and ripened for about five to six months and usually smoked. The average ripening temperatures are 12–16 °C, while the relative humidity is kept at 75–85% [42,47,48].

There are some differences in traditionally and industrially produced *Kulen*. Traditional small-scale household *Kulen* production is driven by cultural practices related to the production region, characterised by uncontrolled and variable production conditions and seasonality (winter months). Contrary to industrial production, home-based production involves no bacteria inoculation and no mould starter cultures. Industrial production, on the other hand, is independent of season and climate changes and takes place in a chamber in which all critical production parameters (temperature, humidity, air flow rate) are regulated automatically. Such production is faster, with a shorter ripening period (of up to three months) due to the use of starter cultures and additives. As said before, industrial production involves no mould starter cultures but does employ lactic acid bacteria and *Staphylococcus*. Industrial ripening chambers are equipped with biological micro-filters and pressure barriers seated at the entrance point that prevent the outdoor air from flowing into the chamber [1,47,48].

### 4.2. Isolation and Traditional Identification of Mycobiota

Upon the delivery of the samples into the laboratory, moulds were isolated by virtue of swabbing *Kulen* surfaces with sterile damp swabs and transferring visible mould colonies onto DG-18 agar (dichloran 18%-glycerol, Merck, Darmstadt, Germany), to be incubated for seven days in darkness at 25 ± 1 °C. Pure mould cultures were sub-cultivated on DG-18 agar, malt extract agar (MEA, BD Difco, Franklin Lakes, NY, USA), and Czapek yeast extract agar (CYA, BD Difco, Franklin Lakes, NY, USA), and incubated for seven days at 25 ± 1 °C in darkness, to be subsequently identified in a traditional manner based on their macro- and micro-morphology according to the published guidelines [30,34].

### 4.3. Molecular Identification of Mycobiota

Mould isolates were additionally identified at the species level using benA (beta-tubulin) and CaM (calmodulin) sequencing [34]. Firstly, the DNA was extracted from isolated mould colonies using the NucleoSpin Microbial DNA Kit (Macherey-Nagel GmbH & Co.KG, Germany) according to the manufacturer’s instructions. Primers specific for benA—Bt2a (5′-GGTAACCAAATCGGTGCTGCTTTC-3′) and Bt2b (5′-ACCCTCAGTGTAGTGACCCTTGGC-3′) & CaM loci—Cmd5 (5′-CCGAGTACAAGGARGCCTTC-3′) and Cmd6 (5′-CCGATRGAGGTCATRACGTGG-3′) were selected for polymerase chain reaction (PCR) amplification. The reaction mixture (25 µL) consisted of 1 µL of the template DNA, 12.5 μL of the PCR buffer (GoTaq G2 Hot Start Green Master Mix Kit, Promega, Madison, WI, USA), 0.4 µM of each primer, and nuclease-free water. The PCR was performed in a 51149-2 thermal cycler (Prime Thermal Cycler, Staffordshire, UK) under the following cycling conditions: 95 °C for 2 min followed by 40 cycles at 95 °C for 30 s, 56 °C for 30 s, 72 °C for 60 s, concluding with 72 °C for 5 min. The PCR products were checked using gel electrophoresis in 1.5%-agarose gel and visualized by virtue of UV trans-illumination (UVIDOC, UVITEC, Cambridge, UK). After the ExoSAP-IT PCR clean-up reagent (Affymetrix, Santa Clara, CA, USA)-mediated purification, amplicons were sent to a commercial facility for sequencing (Macrogen, Amsterdam, the Netherlands). The sequences were aligned and edited using the DNASTAR Software (Lasergene, Madison, WI, USA) and then compared to those available in the the GenBank database using the BLAST algorithm. Obtained sequences are deposited in GenBank database with accession numbers as follows: OK562708 for *A. pseudoglaucus*, OK562709 for *A. proliferans,* OK562710 for *A. tubingensis,* OK562721 for *A. westerdijkiae,* OK562711 for *P. solitum,* OK562720 for *P. commune,* OK562712 for *P. corylophilum,* OK562713 for *P. citrinum,* OK562714 for *P. polonicum,* OK562716 for *P. sumatraense*, OK562718 for *P. brevicompactum,* OK562719 for *P. nordicum,* OK562715 for *P. nalgiovense.*

### 4.4. Mycotoxin Determination

#### 4.4.1. Standards and Reagents

CPA (Art. No. C 1530) and AFB_1_ (Art. No. A6636) analytical standards were purchased from Sigma-Aldrich (St. Louis, MO, USA). The CPA and AFB_1_ stock solutions were prepared by dissolving 1 mg of the standard in 10 mL of acetonitrile (100 μg/mL). The OTA standard (Art. No. AC227400050) was obtained from LGC Standards (Wesel, Germany) dissolved in acetonitrile (10 μg/mL). The working solutions were prepared on the day of the analysis. Acetic acid, methanol, and acetonitrile were obtained from Honneywell (Charlotte, NC, USA). Solid reagents and Tween 20 were obtained from Sigma-Aldrich (St. Louis, MO, USA).

#### 4.4.2. Sample Preparation

For the sake of mycotoxin analysis, after removal (peeling) of the casing, *Kulen* samples were firstly chopped with a knife into smaller pieces and then completely homogenized on a Grindomix GM 200 laboratory homogenizer (Retsch, Haan, Germany). The preparation of samples for the CPA analysis was described in detail by [20] and involves the extraction with 25% acetic acid & acetonitrile and the use of rOQ QuEChERS Extraction Packets (Phenomenex, Torrance, CA, USA) and Captiva EMR-Lipid SPE columns (Agilent Technologies, Santa Clara, CA, USA). The preparation of samples intended for AFB_1_ and OTA analysis involved the use of highly specific immunoaffinity columns (AFLA OCHRAPREP^®^, R-Biopharm Rhône Ltd., Glasgow, Scotland) and was carried out according to the manufacturer’s instructions. After the extraction of mycotoxins using 20 mL of 80%-methanol, vortexing, and centrifugation, the samples were filtered and diluted with 0.1% Tween phosphate buffer (PBS) pH = 7.4 (12 mL of the filtered sample with 60 mL 0.1% Tween 20 PBS) and passed through the immunoaffinity columns. The column was rinsed with 20 mL PBS pH = 7.4, while mycotoxins were eluted by virtue of 1 mL methanol back-flushing and 1 mL ultrapure water washing.

#### 4.4.3. LC-MS/MS Analysis

A high-performance liquid chromatograph (1260 Infinity, Agilent Technologies, Santa Clara, CA, USA) consisting of a degasser, a binary pump, an auto-sampler, and a column compartment, was coupled with a triple quadrupole mass spectrometer (6410 QQQ, Agilent Technologies, Santa Clara, CA, USA). Chromatographic separation of mycotoxins was performed on a 150 × 4.6 mm, 5 µm-particle size Gemini analytical column (Phenomenex, Torrance, CA, USA) coupled with a SecurityGuardTM Cartridges Gemini^®^ C18, 4 × 3.0 mm ID pre-column (Phenomenex, Torrance, CA, USA).

Chromatographic and instrumental AFB1 and OTA mass spectrometry settings were described by [15], and those of the CPA analysis by [20]. For each mycotoxin quantification/confirmation, one precursor and two product ions were monitored, as shown in Table 4.

#### 4.4.4. LC-MS/MS Validation

The LC-MS/MS method was validated according to the Guidance document on the estimation of the Limit of Detection (LOD) and the Limit of Quantification (LOQ) for measurements in the field of contaminants in feed and food [49]. Blank samples (10 replicates) of different dry-fermented sausage samples (five types of sausages in two replicates) were spiked with 0.1 µg/kg of AFB_1_, 0.3 µg/kg of OTA, and 3 µg/kg of CPA for LOD/LOQ determination. The samples were then prepared as described above and analysed. For each batch, a 5-point calibration curve was plotted. The concentration ranges were as follows: OTA, 0.2, 0.4, 2.5, 5, and 5 ng/mL; AFB_1_, 0.05, 0.1, 0.25, 0.5, and 1 ng/mL; CPA, 0.5, 2.5, 10, 15, and 25 ng/mL. LOD and LOQ were determined based on the calibration curves’ slopes and signal abundances of the spiked samples. Linearity was tested within the concentration ranges quoted above, while the recovery was determined by analysing 10 blank samples of different dry-fermented sausage samples spiked with 0.1 µg/kg of AFB_1_, 0.3 µg/kg of OTA, and 3 µg/kg of CPA. While solvent calibration is used, the matrix effect was evaluated by comparing the peak areas of each mycotoxin in the standard solution at 0.25 ng/mL for AFB_1_, 0.40 ng/mL for OTA, and 1 ng/mL for CPA, and the blank matrix post extraction (after sample preparation) spiked at the same level.

### 4.5. Statistical Analysis

Statistical analyses were performed using the SPSS Statistics Software 22.0 (IBM, New York, NY, USA). The results were tested for distribution normality using the Shapiro-Wilks test. In order to determine the statistical significance of the differences in mycological parameters descriptive of industrial and traditional *Kulen* samples, the Mann-Whitney U test was used. Decisions on statistical relevance were made at the significance level of *p* < 0.05.

## Figures and Tables

**Table 1 toxins-13-00798-t001:** Validation parameters obtained for mycotoxins analysed in dry-fermented sausage samples (*n* = 10).

Mycotoxins	LOD (μg/kg)	LOQ (μg/kg)	Recovery (%)	Matrix Effect (%)
OTA	0.18	0.59	119.4	1.81
AFB_1_	0.03	0.11	91.4	6.36
CPA	0.49/2.45 *	1.61/8.07 *	97.52	1.86

LOD—limit of detection, LOQ—limit of quantification; OTA—ochratoxin A; AFB1—aflatoxin B_1_; CPA—cyclopiazonic acid; * Matrix LOD and LOQ with the sample dilution factor of 5.

**Table 4 toxins-13-00798-t004:** Instrumental LC-MS/MS settings according to [15,20].

Analyte	Precursor Ion	Fragmentor Voltage (V)	Product Ions	Collision Energy (eV)
OTA	404	130	357.9	25
			239.0	10
AFB_1_	313.1	170	285.1	23
			269.1	30
CPA	337.2	110	196.3	25
			182.1	20

OTA—ochratoxin A; AFB_1_—aflatoxin B_1_; CPA—cyclopiazonic acid.

## Data Availability

The sequences of mould isolates included in this study are openly available in GenBank with accession numbers as it’s described in the section Materials and Methods.

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
