# Peer review of "Mycobiota and Mycotoxin Contamination of Traditional and Industrial Dry-Fermented Sausage Kulen"

_toxins, 2021, doi:10.3390/toxins13110798_

Round 1
Reviewer 1 Report
In discussions should be clearly presented the novelty of the study, the importance of these results in the risk assessment of mycotoxins from food.
Also the limitations of the studies should be clearly presented.
Author Response
Comment: In discussions should be clearly presented the novelty of the study, the importance of these results in the risk assessment of mycotoxins from food.
Answer: In line with the Referee's comment, the revised Discussion section (subsection 3.3. Mycotoxins in traditional and industrial Kulen) now clearly points out that this study investigated into the contamination of Kulen with mycotoxins most important for meat products, as well as that the study includes the comparison between the concentrations of these mycotoxins in traditional and industrial samples (Lines 192-194). On top of that, at the end of the Discussion section the great importance of investigations into mycotoxin occurrence for the assessment of consumer exposure to these contaminants via meat products has been emphasised, in particular that of the investigations into CPA as a virtually unexplored mycotoxin. It has also been pointed out that the research in this field is of essence for compiling legislation governing mycotoxin presence in meat products, since no maximal or guidance values have insofar been established in European countries for this type of products (Lines 311-317).
Comment: Also the limitations of the studies should be clearly presented.
Answer: It has now been clearly stated in the revised Discussion section that further investigations into the presence of mycotoxin biosynthetic genes in toxigenic species are needed, as well as that the limitation of this study is a smaller number of samples, which were not obtainable in larger quantities due to the limited number of available Croatian Kulen producers, especially industrial ones (Lines 250-253).
The authors are indebted to our esteemed Reviewer for their most helpful suggestions and comments.
Reviewer 2 Report
I appreciate the great effort of authors to conduct this study, in order to identify and compare surface mycobiota of Kulen sausages, and to detect contamination with mycotoxins.
The article has a great importance, due to the fact that "OTA was detected in 27% of the analysed Kulen samples, out of which 38% were 226 traditional Kulen samples in which OTA concentration of up to 6.95 µg/kg was found ".
The situation is even worse given the fact that there is no guidance value for OTA or other mycotoxins in meat and meat products, at the European level.
I don't have many comments, but I want to ask a few questions:
1.Are all studied sausages made with natural membrane, pig appendix (intestinum caecum)?
2.Were the samples for micotoxins analysis taken from inside of the products or from their surface, from the membrane? I ask this because in Abstract and from the title of table 2 it is clear that the samples were taken "to identify and compare mycobiota surface……..", but then to 2.4.2. Sample preparation, starting with line 404, it is not specified from which part of the product the samples are taken for micotoxins determination.
3.Line 357 – "Staphylococci" should be replaced by Staphylococcus.
Author Response
Comment: I appreciate the great effort of authors to conduct this study, in order to identify and compare surface mycobiota of Kulen sausages, and to detect contamination with mycotoxins. The article has a great importance, due to the fact that "OTA was detected in 27% of the analysed Kulen samples, out of which 38% were 226 traditional Kulen samples in which OTA concentration of up to 6.95 µg/kg was found ". The situation is even worse given the fact that there is no guidance value for OTA or other mycotoxins in meat and meat products, at the European level. I don't have many comments, but I want to ask a few questions: Are all studied sausages made with natural membrane, pig appendix (intestinum caecum)?
Answer: In line with the recipe and the valid Regulation, Kulen is produced solely using a natural membrane – the pig appendix (intestinum caecum), so that all samples under this study, both traditional and industrial, were produced accordingly, as described in Lines 348-356.
Comment: Were the samples for micotoxins analysis taken from inside of the products or from their surface, from the membrane? I ask this because in Abstract and from the title of table 2 it is clear that the samples were taken "to identify and compare mycobiota surface…", but then to 2.4.2. Sample preparation, starting with line 404, it is not specified from which part of the product the samples are taken for micotoxins determination.
Answer: We are grateful to the Referee for spotting the omission in the methodology description. To make amends, the revised subsection 4.4.2. Sample preparation now brings the information that mycotoxin analysis was preceded by the removal (peeling) of the Kulen casing. The samples were then chopped with a knife into smaller pieces and completely homogenized using a laboratory homogenizer (Lines 416-417).
Comment: Line 357 – "Staphylococci" should be replaced by Staphylococcus.
Answer: The requested replacement has been made (Line 367).
The authors are indebted to our esteemed Reviewer for their most helpful suggestions and comments.
Reviewer 3 Report
Manuscript Number: toxins-1450487
Title: Mycobiota and mycotoxin contamination of traditional and industrial dry-fermented sausage Kulen.
Type of manuscript: Article
Major comments:
1. The subject of the manuscript is consistent with the scope of TOXINS special Issue: "Mycotoxin Contamination and Food Safety".
2. There are no errors of fact or logic.
3. The abstract does bring out the main points of the paper.
4. The information obtained from the study was processed and presented clearly and concisely.
5. Basic validation parameters of LC-MS/MS method are described eg LOD, LOQ and recovery.
6. Page 5, Table 3. Mycotoxin concentrations in traditional and industrial Kulen
… Mean of positives* *mycotoxin concentration above the LOD …
What about left censored data approach (using lower bound (LB), middle bound (MB) and upper bound (UB)?
Can you explain that?
7. The literature references are adequate and recent.
8. Manuscript describing research involving dry-fermented sausage Kulen is only local interest?
Nevertheless, I believe that it brings new knowledge in the field of dry-fermented sausage contamination by toxigenic microfungi and mycotoxins in Europe. It is interesting to compare the results of traditional Kulen samples with industrial Kulen samples.
Minor edits:
1. Page 4, line 146.
Instead … (4 - 43°C)… specify (4 - 43 °C)
2. Page 4, line 151.
Instead … 5 oC … specify 5 oC
3. Page 5, Table 3. Mycotoxin concentrations in traditional and industrial Kulen
Instead … % / N of positives* … specify n+: number of positive samples and n+%: per cent of positive samples.
4. Page 7, line 304-307.
Instead … products using highly sensitive analytical methods (Bailly and Guerre, 2009; Bailly et al., 2005; Zadravec et al., 2020; Lešić et al., 2020). Bailly et al. (2005) reported relatively high amounts of citrinin produced in dry meat by toxigenic P. citrinum strain after 16-day incubation at 20 °C…
Specify … products using highly sensitive analytical methods [00-00]. Bailly et al. (2005) reported relatively high amounts of citrinin produced in dry meat by toxigenic P. citrinum strain after 16-day incubation at 20 °C [00]. …
So, the manuscript is fit for possible publication in TOXINS journal (Accept after minor revision)!
Author Response
Comment: Page 5, Table 3. Mycotoxin concentrations in traditional and industrial Kulen…Mean of positives* *mycotoxin concentration above the LOD…What about left censored data approach (using lower bound (LB), middle bound (MB) and upper bound (UB)? Can you explain that?
Answer: We agree that the left censoring data approach is important for risk assessment aiming at the estimation of population exposure to mycotoxins as food contaminants. However, the aim of this study was to investigate into the occurrence of mycotoxins in one of the important types of traditional meat products and to compare their presence in traditional and industrial samples. The output of this research, together with other data on mycotoxin prevalence in various meat products, can be of further use for the well-founded risk assessment and the assessment of consumer exposure to these contaminants via meat products’ consumption. For this very reason, within the frame of this study we chose to use the common manner of study results’ presentation, most often resorted to by other study authors investigating into the presence of other food contaminants, as well. Another reason for not using the left censoring data approach were low LOD values descriptive of AFB1 and OTA presence (the latter being tracked only in form of residues) and the fact that AFB1 was not detected in any of the study samples. Of course, should our esteemed Referee still be of the opinion that such a presentation would add value to this research and can be fit into Table 3, we can do it by all means. In that case, the lower bound (LB) and the upper bound (UB) can be obtained according to the “Management of left-censored data in dietary exposure assessment of chemical substances” (EFSA, 2010). This approach implies that the LB is obtained by assigning the zero (the minimum possible) value to all samples reported as lower than the LOD. The UB can be obtained by assigning the numerical value of LOD to the values reported as < LOD.
Comment: Manuscript describing research involving dry-fermented sausage Kulen is only local interest? Nevertheless, I believe that it brings new knowledge in the field of dry-fermented sausage contamination by toxigenic microfungi and mycotoxins in Europe. It is interesting to compare the results of traditional Kulen samples with industrial Kulen samples.
Answer: We are most obliged to the Referee for the recognition and appreciation of the significance of this research. The aim of the study is defined exactly as stated by the Referee, with a special emphasis on the need for investigation into the still under-investigated CPA mycotoxin, which was detected in meat products even in substantial concentrations.
Comment: Page 4, line 146. Instead… (4 - 43°C)… specify (4 - 43 °C)
Answer: The required specification has been made (Line 144).
Comment: Page 4, line 151. Instead… 5oC … specify 5 oC
Answer: The required specification has been made (Line 149).
Comment: Page 5, Table 3. Mycotoxin concentrations in traditional and industrial Kulen. Instead … % / N of positives* … specify n+: number of positive samples and n+%: per cent of positive samples.
Answer: The markings in Table 3 have been changed in line with the Referee’s suggestions.
Comment: Page 7, line 304-307. Instead … products using highly sensitive analytical methods (Bailly and Guerre, 2009; Bailly et al., 2005; Zadravec et al., 2020; Lešić et al., 2020). Bailly et al. (2005) reported relatively high amounts of citrinin produced in dry meat by toxigenic P. citrinum strain after 16-day incubation at 20 °C…Specify…products using highly sensitive analytical methods [00-00]. Bailly et al. (2005) reported relatively high amounts of citrinin produced in dry meat by toxigenic P. citrinum strain after 16-day incubation at 20 °C [00]. …
Answer: In the revised manuscript, all references have been quoted according to the Instructions for Authors. The references elaborated by the Referee have now been numerated in the body text (Lines 307-308).
In the revised version, changes to the original manuscript are marked using the Track Changes option. The authors are indebted to our esteemed Reviewer for their most helpful suggestions and comments.
Round 2
Reviewer 1 Report
The authors addressed all my comments. The manuscript is ready for acceptance.